# A Novel Inexpensive Camera-Based Photoelectric Barrier System for Accurate Flying Sprint Time Measurement

**DOI:** 10.3390/s23177339

**Published:** 2023-08-23

**Authors:** Tom Uhlmann, Sabrina Bräuer, Falk Zaumseil, Guido Brunnett

**Affiliations:** 1Faculty of Computer Science, Chemnitz University of Technology, Straße der Nationen 62, 09111 Chemnitz, Germany; 2Faculty of Behavioral and Social Sciences, Institute of Human Movement Science and Health, Chemnitz University of Technology, Reichenhainer Straße 31-33, 09126 Chemnitz, Germany

**Keywords:** photoelectric barriers, sprint time measurement, sports performance measurement, mobile device

## Abstract

This paper introduces a novel approach to addressing the challenge of accurately timing short distance runs, a critical aspect in the assessment of athletic performance. Electronic photoelectric barriers, although recognized for their dependability and accuracy, have remained largely inaccessible to non-professional athletes and smaller sport clubs due to their high costs. A comprehensive review of existing timing systems reveals that claimed accuracies beyond 30 ms lack experimental validation across most available systems. To bridge this gap, a mobile, camera-based timing system is proposed, capitalizing on consumer-grade electronics and smartphones to provide an affordable and easily accessible alternative. By leveraging readily available hardware components, the construction of the proposed system is detailed, ensuring its cost-effectiveness and simplicity. Experiments involving track and field athletes demonstrate the proficiency of the proposed system in accurately timing short distance sprints. Comparative assessments against a professional photoelectric cells timing system reveal a remarkable accuracy of 62 ms, firmly establishing the reliability and effectiveness of the proposed system. This finding places the camera-based approach on par with existing commercial systems, thereby offering non-professional athletes and smaller sport clubs an affordable means to achieve accurate timing. In an effort to foster further research and development, open access to the device’s schematics and software is provided. This accessibility encourages collaboration and innovation in the pursuit of enhanced performance assessment tools for athletes.

## 1. Introduction

To precisely evaluate an athlete’s performance, it is crucial to establish objective measurements. These measurements encompass a range of performance parameters, such as strength, agility, speed, and acceleration. To assess each property, test batteries are employed, and their efficacy depends on meeting fundamental criteria, including reliability, accuracy, and comparability, ensuring consistent and dependable outcomes. While some parameters are relatively easy to assess, such as strength through the number of push-ups or pull-ups, others pose challenges in accurate measurement, like speed and acceleration in sprints. For sports where maximum running speed over a short distance is crucial, precise and reproducible timing is essential. This involves measuring the time for a single repetition, such as a 30 m sprint, or multiple repetitions, like a pendulum run. The importance of accuracy in such measurements cannot be overstated, as even fractions of a second over the course of 30 m can signify improvements or declines in performance. Consequently, electronic timing devices are widely employed due to their superior accuracy compared to manually operated stopwatches. Additionally, reproducibility is crucial, as measurements will be repeated to track an athlete’s performance changes. The timing system should remain unaffected by external conditions, ensuring it is solely influenced by the athlete’s performance, without interference from setup errors or human mistakes.

This study endeavors to conduct a comprehensive review of the latest technologies and commercially available devices utilized for timing short distance sprints. It will take into account their accuracy and pricing, aiding in making informed decisions about which system to use for training sessions, seeking accurate measurements with minimal effort and expenditure.

An increasing number of coaches are turning to smart devices like smartphones and tablets to assess athletes’ performances [1]. These devices hold great potential as valuable tools for athletes and coaches, given their widespread availability and built-in array of features and sensors. However, only a few options to time short distance sprints are available. This research will leverage image evaluation techniques to develop a system akin to photoelectric cells timing systems, utilizing consumer-grade camera hardware found in every smartphone. The objective is to present a novel, cost-effective, and open-source timing system that rivals the accuracy of commercial devices while being more accessible and affordable, particularly benefiting smaller sport clubs and schools. The proposed system will undergo an evaluation in comparison to a premium commercial photoelectric barrier system.

Specifically, this research provides the following contributions:Extensiv review of existing system to time short distance runs, including apps for smart devices.Proposal of an algorithm that simulates photoelectric barriers in a video stream akin to photoelectric cells.Development of a low-cost timing system for short distance sprints using low-cost consumer-grade electronics and an Android device as well as evaluating its accuracy through two real-condition experiments.Open-source publication of the device’s schematics and software.

The remainder of this paper is structured as follows: The next section will delve into the technologies used and available devices for timing short distance sprints. It will also summarize the findings regarding the accuracy observed in practical applications and its implications for assessing athlete performance. Section 3 will detail the innovative approach of simulating photoelectric barriers within a video stream and introduce the timing device built upon this principle. Following that, Section 4 will present the evaluation of the device’s accuracy. In the subsequent Section 5, we will discuss the results and explore potential future research directions.

## 2. Time Measurement Techniques for Short Distance Sprints

The field of sports science has witnessed significant advancements in technology, leading to the availability of numerous commercial systems for timing short distance sprints. These systems play a pivotal role in accurately measuring athletes’ performance and providing invaluable insights to coaches, athletes, and researchers alike. A timing system typically comprises one or several gates, acting as measurement lines where sensors detect athletes passing by. There are various techniques and systems available for timing short distance sprints and pendulum runs, including reflective photoelectric cells (RPC), magnetic sensors, GPS combined with an Inertial Measurement Unit (IMU), Radar, and camera image evaluation. The upcoming subsections will review each technology’s use for timing and assess their accuracy based on scientific literature experiments. Table 1 presents an overview of available commercial devices, including their prices, utilized technology, accuracy, and whether they function as closed systems or require additional mobile devices. The next sections will delve into the workings of these techniques, highlighting their respective benefits, limitations, and applications in timing short distance runs. The accuracy of these systems will also be addressed, considering the difference between each system and a reference photoelectric cell system, unless otherwise stated.

### 2.1. Stopwatch

Stopwatches serve as the most basic timing devices and are operated manually by the user. To begin the measurement cycle, the operator simply presses a button, and to stop, they press it again. This simplicity allows for taking intermediate or multiple times with ease, making them highly accessible due to their small size that can fit into any pocket. However, stopwatches have some drawbacks, including relatively low accuracy, which is heavily reliant on the operator’s skill and experience, leading to a wide spread of results.

In a study by Rodríguez et al. [2], they measured athletes’ performance in 4 times 10-m shuttle sprints and 30-m dashes using a photoelectric cell timing system. The reported times were then compared against those recorded by trained and untrained stopwatch operators. Their findings suggested that the accuracy for shuttle runs was better than 350 ms and for the 30 m dashes better than 500 ms. The study also noted that trained operators achieved slightly more accurate results, though the effect was marginal. Hetzler et al. [20] conducted a comparison between timings of 200-m runs using an electronic timing system and handheld stopwatches. Although they found no significant difference in the mean error, the reported absolute error for handheld stopwatches was 0.15±0.20 and 0.16±0.19 s for the two measurement methods they used. The accuracy of hand-held stopwatches appears to vary significantly across different studies. However, by averaging the results from various research, an approximated accuracy of 350 ms can be derived, which will serve as a reasonable and sufficient estimate for comparison purposes. There is a wide range of commercial handheld stopwatch devices available, often at a low cost of just a few euros. Additionally, nearly every smartphone is equipped with an integrated stopwatch app, making them easily accessible. However, it should be noted that physical button-operated stopwatches generally provide more accurate results compared to touch display-based alternatives.

### 2.2. Reflective Photoelectric Cell Timing Systems

Reflective photoelectric cells (RPC) are widely adopted in time measurement for short distance sprints due to their accuracy and effectiveness. These systems use infrared or laser beams emitted on one side and reflected back by a reflector on the opposite side, creating a gate typically spanning 1 to 3 m. When the light beam is obstructed by the athlete, the barrier is assumed to be crossed. High-class RPC systems employ a dual barrier setup with two beam-reflector pairs spaced several centimeters apart laterally. This ensures that the gate is considered crossed only if both beams are obstructed, preventing the possibility of an arm or leg triggering the barrier before the athlete’s upper body, thereby enhancing accuracy.

The extremely high sampling rate of the sensor, allowing reaction times of 1 ms or even less, enables almost instantaneous detection of obstacles, contributing to the high accuracy of RPC systems. They have been used widely and are established as the gold standard for time measurement in short distance sprints. Bond et al. [4] conducted a study comparing RPC systems and video-based systems against 3D motion capturing in short distance sprints. They found that RPC system accuracy depends on the sensor’s height, with best results achieved when placed at the athlete’s hip height. The study reported video capture system accuracy at about 10 to 20 ms and photocell accuracy at about 30 ms. They assumed the 3D motion capturing system to be the most accurate.

While RPC systems offer remarkable accuracy, the setup in a mobile setting can be time-consuming. Aligning the transceiver and reflector often requires a frustrating trial and error process, and the lack of wireless connectivity to the control device further increases setup times. Nevertheless, these devices strike a favorable balance between cost, accuracy, and ease of use, making them the most widely used devices for timing runners.

In the market, several devices utilizing this technique are available, and three such devices with public pricing information are listed in Table 1. Among them, the Cronox 3.0 stands out as it shares a similar setup to the system proposed in this paper, albeit being one order of magnitude more expensive. With their proven track record, RPC systems continue to be a reliable choice for timing short distance sprints, offering a practical and efficient solution for athletes, coaches, and researchers.

### 2.3. Magnetic Sensors Timing Systems

Magnetometers, as sensors capable of detecting the strength of nearby magnetic fields, serve as the foundation for a fairly new timing technique. To create a gate, one or multiple bar magnets are placed along the line to be measured. The timing system will assume the athlete passes this gate, when the peak value of the magnetic field reported by the sensor occurs. One major advantage of this technique is its ease of use once installed. Athletes can be equipped with small active sensors, and contemporary smartphones equipped with magnetometers can be utilized for timing as well. However, there are notable drawbacks. Integrating magnets into the track requires careful installation and results in the gates being fixed. Additionally, athletes wearing sensors might find them uncomfortable, potentially impacting their performance.

Fasel et al. [21] proposed such a magnet-based timing system for alpine skiing, utilizing bar magnets to create the gate. They reported an accuracy of 25 ms for gate crossings. Similarly, Buxade et al. [8] evaluated a magnetic system using bar magnets and the magnetometer inside an Inertial Measurement Unit (IMU) in 60 m sprint running tests and ski-slalom. Their results showed accuracies of 77 ms and 50 ms, respectively.

Currently, the company SmarTracks [7] offers a commercial system using magnetic sensors for tracks. Once installed, it can be used for free with just a Smartphone or the company’s DX05 sensor. According to the company, the accuracy of the system is in the millisecond spectrum. For this review, an accuracy of 77 ms is assumed based on the study by Buxade et al. [8], as it is the only one utilizing a similar system for short distance sprints. SmarTracks also offers a mobile version of their system, but there is no data available regarding its accuracy or cost.

### 2.4. RFID Timing Systems

Radio frequency identification (RFID) chips offer wireless data transfer capabilities. To enable this, special antennas are required at the gate to read information from the RFID chip. Unlike magnetic sensor timing systems, athletes using RFID wear passive sensors, which can be as small as a stamp since they don’t require a power supply. However, the installed system must be active, making it challenging to set up in a mobile fashion and expensive in the upper four-figure range [9]. RFID systems find extensive use in mass sport events like 5 k runs or marathons, where they assess the running time for each participant.

Woellik et al. [10] evaluated an RFID timing system integrated into a track and field stadium. They reported an average error of less than 25 ms. Their data suggests an accuracy of more than 45 ms for a single chip, better than 35 ms for two chips, and more than 30 ms for four chips used on a single athlete. While the used RFID chips [22] may have a response time of 3 ms, the actual accuracy is lower by an order of magnitude. This shows that it is crucial to distinguish between time resolution of the sensor and actual accuracy in this context.

Operating such systems also requires significant training, and computer systems are necessary to run them efficiently. The drawbacks of massive upfront costs and the need for proper training for operation highlight the complexity and investment associated with RFID timing systems. However, they continue to be instrumental in accurately capturing running times for mass sports events, making them indispensable tools for large-scale competitions. There is no widespread use of such systems for short distance sprint timing.

### 2.5. GPS-Based Monitoring

Electronic Performance and Tracking Systems (EPTS) have become a mainstay in professional team sports during training. These systems collect an extensive array of athlete data, including running speed, distance covered, position on the field, and heart rate. With the help of sophisticated algorithms, this data can be harnessed to assess an athlete’s performance and also identify signs of potential injuries caused by training overload. Typically, these systems comprise a pack of sensors, including GPS, Inertial Measurement Unit (IMU), and heart rate sensor, all embedded in a vest worn by the athlete during training sessions. The sensors continuously collect data, which is transmitted to the system for storage and evaluation.

Although highly complex and immensely beneficial for team sports, EPTS are not designed to accurately measure short distance sprints. Furthermore, operating these systems requires a team of professionals, and little data is available regarding their accuracy for short distance sprint scenarios. Additionally, GPS sensors work effectively only outdoors, and their accuracy is subject to various limitations. The GPS sensors’ usual operating rate of 10 Hz yields a theoretical maximum accuracy of 100 ms. Single frequency GPS systems, prevalent in consumer devices, exhibit average accuracy to the order of 4.9 m [23] under open sky conditions. The accuracy deteriorates under cloudy skies, near bridges, or dense tree cover, making the measurements less accurate and reproducible. Although dual frequency GPS systems offer centimeter-level accuracy, their size and costs restrict their usage to professional and military applications. There is potential to enhance accuracy through sensor fusion, as IMUs can estimate running parameters such as speed [24]. However, commercial systems’ proprietary algorithms make it challenging to ascertain their data calculation methods.

Research evaluating GPS timing systems has been conducted, comparing them to electronic timing gates [25], radar guns [26], video-based analysis [14], and radar gun setups [27] in shuttle run settings. These studies demonstrate that such GPS systems exhibit an accuracy of about 2 km/h, similar to radar guns. However, given the low sampling rate, limited positioning accuracy, and susceptibility to atmospheric conditions, GPS systems may not be reliable sources of information for short distance sprints.

Unfortunately, data on the accuracy of GPS-based systems listed in Table 1 is not available, preventing estimation of their accuracy for short distance sprint scenarios. In light of this limitation, further research and development are required to address the specific challenges of timing short distance sprints using this technology.

### 2.6. Radar-Based Speed Measurement

Radar guns utilize the Doppler effect to measure the speed of a moving object. This effect refers to the change in frequency of sound or light waves when the source of the waves is in motion relative to the observer. When a radar gun emits a beam of radio waves, they reflect back to the gun after striking the object. The frequency of the reflected waves differs from that of the emitted waves due to the object’s motion towards or away from the gun. This change in frequency is directly proportional to the object’s speed, allowing the radar gun to calculate it. The speed is then displayed on a digital readout on the gun.

Radar guns are commonly known for their use by law enforcement officers to assess vehicle speed. Additionally, they find application in some sports leagues to measure the maximum speed of athletes or projectiles, such as baseballs or footballs. The accuracy of hand-held radar guns, according to manufacturer claims, falls within the range of about 2 km/h. However, there are limitations to the use of radar guns for timing short distance runs. These devices require trained and licensed operators to accurately predict vehicle speed, which implies that untrained operators will achieve lower accuracy rates. Moreover, radar guns are designed to measure only objects moving faster than 15 km/h (approximately 4.5 m/s), a speed already considered fast for a human. When considering the additional error margin of several km/h, precise speed assessment of running athletes becomes unfeasible. Furthermore, radar guns can solely assess the speed of an object and not its position, necessitating the use of additional sensors to time a short distance run. Considering these limitations, radar guns are not a viable option for accurately timing this scenario.

### 2.7. Fully Automatic Photo Finish Timing

In order to precisely measure an athlete’s finishing time in competitive sports like track and field, horse racing, canoeing, or cycling, photo finish systems have been used for a long time. At the finish line, a strip photo is taken to capture a two-dimensional image with the finish line in one dimension and time in the other. This enables to identify the precise moment the athlete crosses the finish line. The specific attribute of the athlete that determines whether the line has been crossed heavily depends on the kind of sport. Automatic evaluation is challenging because of this. For instance, in track and field, the athlete’s shoulder needs to pass the finish line. Image change detection techniques were used by Zaho et al. [28] to remove the background and segment the athlete into various regions. Then, they were able to estimate the times of track and field athletes’ finishes with an accuracy of 4 ms and a precision of 86.3 percent. The method was improved by Li et al. [29], who reported a 2 ms accuracy, but with a slight decrease in correctness. To provide a time resolution of 1 ms and sharp images, strip photographs require high FPS line scan cameras that take at least 1000 images per second. Such devices cannot be regarded as inexpensive because they already cost several thousand euros.

Manual inspection of an athlete crossing the finish line allows for the precise and accurate measurement of their time, making these the most accurate timing systems currently available. However, this level of accuracy comes with certain trade-offs. These systems necessitate extensive setup time, involving several trained operators, and the costs involved typically fall within the lower to mid 5-figure range. As a result, these sophisticated timing systems are primarily reserved for use in official competitions rather than as a practical economic option for regular training sessions. While they offer unparalleled accuracy, the associated setup and operational requirements limit their feasibility for routine training scenarios. Nevertheless, in competitive environments where precision is paramount, manual assessment of the strip photo remains the gold standard, ensuring fair and accurate results for athletes and spectators alike.

### 2.8. Camera-Based Evaluation Techniques

Contemporary cameras are inexpensive but still capable of producing high-quality photos and videos. Since every smartphone has at least one camera, they are widely accessible. Because of this, the research community has worked hard to replace expensive equipment with equally reliable video-based analysis techniques. For instance, frame-by-frame playback, video comparison, as well as manual and automatic annotations, are all features of the open source program Kinovea [30]. A high FPS video camera that can capture more than 100 images per second is usually required for the majority of analysis techniques. Lower capture rates frequently have slower shutter speeds and lower resolution in time, which causes strong motion blur when capturing fast movements. If the necessary features cannot be observed clearly, analysis and assessment become challenging or impossible. Additionally, difficult lighting situations, like low light settings, can significantly lower the quality of videos. Despite these drawbacks, camera-based methods have already demonstrated their ability to replace expensive IR-Systems or force plates. Balsalobre-Fernandez et al. [31] used a consumer grade high speed camera to capture and assess the performance of counter movement jumps. They recorded jumps at 240 FPS and used Kinovea to evaluate the measurement data. They achieved results on par with professional IR-based system, although with the drawback of manual annotation of the videos, which took about 30 s for each clip.

Camera-based systems for timing short distance sprints have shown their viability through the use of strip photography. However, this subsection will concentrate on existing systems and research employing consumer-grade cameras, such as those found in contemporary smartphones. These cameras offer the advantage of on-site video recording and real-time analysis, making them accessible and convenient for various applications.

There are numerous apps available for automated or semi-automated performance measurement, motion tracking, and analysis. For a summary of these apps, we would like to direct the reader to Busca et al. [1]. It should be noted that the majority of them are only available for Apple devices. The fact that these smartphones have much more capable camera hardware than the majority of Android devices could be the cause. While the majority of low-cost and middle-class mobile devices can only record videos at a frame rate of 30, many iPhone models offer capture rates of 120 or even 240 FPS. Only expensive, high-end Android devices come with comparable features. Because of their blurry images or poor time resolution, the aforementioned apps would therefore not be very useful to the majority of Android users.

The My Sprint app [16] is a specialized tool designed explicitly for evaluating short distance sprints, and its scientific validity has been assessed by Romero-Franco et al. [17]. The study involved comparing short distance sprints of 40 m using three different timing methods: photocell timing, radar gun system, and the My Sprint iOS app. Additionally, the app computed a series of sprint-performance parameters based on a simple method for measuring power force [32]. The video recordings were captured at a high frame rate of 240 fps at 720 p resolution. For accurate timing, the camera must be positioned in the frontal plane to capture the athlete’s full run, which in this case covers 18 m to register the entire sprint. To account for parallax, the setup required sophisticated placement of markers. The main drawback of the My Sprint app approach is that it lacks automatic detection of the specific frame where the athlete’s hip crosses each marker. This manual selection of frames for each run and marker can be time-consuming and introduces some level of subjectivity. However, the study presented the Bland-Altman plot, which demonstrated that the accuracy of the My Sprint app is approximately 28 ms.

The SprintTimer app [18] is a user-friendly tool that offers strip photography on iOS. However, it requires a device with a high-speed camera feature to produce strip photos of sufficient quality. Without this feature, swift movements may result in blurry images, making it challenging to distinguish the required features in the photograph. Despite this limitation, the SprintTimer app has gained popularity among hobbyists and small sport clubs. It provides an easy-to-use solution for timing sprints and other events, generating a strip photo directly on the phone. Automatic detection features are not present in the system, necessitating manual examination of the resulting strip photo to obtain the timing information. The system is not designed to time flying runs. Regarding accuracy claims, the app’s developer asserts an accuracy of 10 ms. To support this claim, he constructed an artificial test rig using a pendulum. Bland-Altman plots displayed in a blog post [33] demonstrated accuracy well within the claimed 10 ms range, though accuracy may vary between different mobile devices. The app’s timing results have not been compared with other timings systems in a realistic practical scenario or published in a peer-reviewed manner.

An app available for Android devices is Photo Finish [19]. This app utilizes an algorithm to recognize athletes passing by from the recorded video, with a particular focus on detecting the athlete’s chest line to precisely determine when the capture line was crossed. For flying sprints, which require timing using multiple devices, the app requires several smartphones, WiFi, or mobile internet, and the paid version. Being a closed-source commercial application, specific details about its functioning are limited. According to the product page, the software captures 30 full-screen images per second and identifies the chest in each one. The precise time is then interpolated from the two images taken just before and just after the finish line. The developers of Photo Finish claim an accuracy of 10 ms, but no comparison to existing systems has been conducted yet. Additionally, it remains uncertain how the chest detection feature works and how reliable it is, particularly under low-light conditions. Overall, Photo Finish provides a unique solution for Android users seeking to time athletes using a chest detection algorithm. While its claimed accuracy is promising, further research and comparisons with other established systems would be beneficial to validate its performance. Additionally, more information about the chest detection functionality would instill greater confidence in its reliability, especially in challenging lighting scenarios.

### 2.9. Literature Summary and Evaluation

The field of timing short distance sprints encompasses a wide array of systems utilizing various technologies, each with its own set of requirements, advantages, and drawbacks. A thorough review of the scientific literature reveals that a higher price tag does not necessarily translate to higher accuracy in these systems regarding short distance sprints. Most of the existing systems boast an accuracy within the range of 30 to 50 ms. Interestingly, the “gold-standard” reflective photoelectric cell (RPC) systems, commonly used for comparison, themselves exhibit an accuracy of 30 ms. The sole exception to this range are the Photo-Finish systems, which employ high-fps line scan cameras and manual evaluation of the strip photo, enabling them to achieve an impressive 1 ms accuracy. However, existing studies do not confirm fully automatic electronic timing systems with accuracies better than 30 ms.

It can be concluded that differences of 50 ms in timing are well within the limits of the existing automated commercial systems. This consideration should be kept in mind when comparing and ranking athletes’ performances, especially when their times were measured using different devices or at different locations. Despite this, the use of electronic timing systems remains crucial for assessing short distance running speed, as it provides accuracy one order of magnitude higher than manually operated stopwatches. The widespread availability of smartphones and tablets, equipped with an array of sensors, has paved the way for cost-effective solutions comparable to expensive commercial systems. This has made electronic timing accessible even to small sport clubs and schools, although most solutions are currently limited to high-end mobile phones or require the use of multiple devices. In this context, this paper aims to explore the effectiveness and accuracy of low-cost consumer-grade camera systems for timing short distance sprints. Such research holds promise for providing affordable and reliable solutions for timing athletes’ performances, further advancing the field of sports timing and analysis.

## 3. The Novel Photoelectric Virtual Barrier System

### 3.1. Virtual Image-Based Photoelectric Barriers

This section will first describe our image-based photoelectric barrier technique for motion detection in front of a camera, followed by a description of its practical implementation. Since microprocessors are the hardware we are targeting, we must employ a method that is computationally efficient and affordable to ensure a constant, high FPS and maximum precision. As a result, we will base our strategy on the most basic image change detection technique and use the mean squared error (MSE) metric to compute the color channel differences between two subsequent images.

Image change detection is a well-studied topic with applications in a variety of industries, including video surveillance, medical diagnosis, monitoring civil infrastructure, driver assistance systems, and remote sensing, which focuses primarily on the analysis of satellite images. The task is to identify areas of change within two images of the same scene that were taken at different times. Over the past few decades, a wide range of methodologies have been proposed for use in various contexts where automatic image evaluation is advantageous or necessary. Radke et al. [34] compiled a thorough review of these methods. We use the simplest and most computationally efficient type, the simple differencing approach, and demonstrate how to address the limitations because the advanced approaches are computationally demanding and cannot be used on the targeted hardware.

Due to the memory, bandwidth, and processing power restrictions of microcontrollers, more sophisticated techniques are not practical. This would go against the need to complete the calculations in a split second. Since only the detection of any motion in front of the device is necessary, and the detection of a specific object is not required, this approach becomes viable.These can be referred to als virtual photoelectric barriers because they behave in a manner that resembles that of conventional photoelectric barriers. Although a large area is covered by a camera’s image, motion detection is only required within a narrow band of the image. Therefore, just a few columns in the middle of the image are defined as the detection area. The region being considered is denoted by the black bar in the leftmost image of Figure 1. With this configuration, the gate encompasses the entire vertical area along with a small horizontal portion in front of the camera. The method can be employed on any device equipped with image processing capabilities, even though its primary development was intended for microprocessors. Consequently, implementation on a smartphone, tablet, or computer is also feasible.

#### 3.1.1. Mathematical Description

Considering a tuple of images $\left\{I_{t},I_{t+1}\right\}$ where $t\in Z^{+}$ denotes the point in time at which the image was captured. The image maps pixel coordinates $x\in R^{2}$ to pixel values $I\left(x\right)\in R^{k}$ with k=1 for grayscale images, and k=3 for RGB colored ones. The region within image *I* where motion detection is intended, denoted as Ω⊆I, encompasses *n* pixels. The image-to-image deviation between two consecutive images is calculated utilizing the MSE metric as described in Equation (Equation 1), where the superscripts *r*, *g*, and *b* refer to the red, green, and blue channels, respectively. For grayscale images, the image-to-image deviation can be computed by considering only a single channel. The choice of the MSE metric is based on its invariance to the size of Ω and its sensitivity to outliers, both contributing to heightened overall sensitivity of the virtual photoelectric barrier.
(1)$$MSE\left(I_{t},I_{t+1}\right)=\frac{1}{n}\sum_{y\in \Omega }\left[{\left(I_{t}^{r}\left(y\right)-I_{t+1}^{r}\left(y\right)\right)}^{2}+{\left(I_{t}^{g}\left(y\right)-I_{t+1}^{g}\left(y\right)\right)}^{2}+{\left(I_{t}^{b}\left(y\right)-I_{t+1}^{b}\left(y\right)\right)}^{2}\right]$$

The decision function (Equation 2) is defined to decide for two consecutive images, whether the difference in the images is sufficiently large to observe motion. The threshold θ∈R is a user-defined parameter that determines the sensitivity of the photoelectric barrier, and τ is the noise level. Both values can be adjusted to increase the reliability of the system in difficult situations. How to estimate reliable parameters is covered in the next subsection.
(2)$$B\left(I_{t},I_{t+1}\right)=\left\{\begin{array}{cc} 1, & MSE(I_{t},I_{t+1})>\theta +\tau \\ 0, & \mathrm{otherwise} \end{array}\right.$$

Equation (Equation 1) can be evaluated by a microprocessor several times per second for a reasonable size of Ω. The metric is quite simple but feasible for this setting, since only a binary decision has to be made (movement yes or no).

#### 3.1.2. Reliability Consideration

As a passive sensor, a camera necessitates consideration of various external factors influencing the accuracy of detection. Given the utilization of a straightforward error metric, the technique is vulnerable to false positives that cannot be resolved algorithmically. These false positives may originate from unintended camera shifts due to wind or ground vibrations, internal camera noise, challenging lighting environments, or an unstable background. To guarantee precise detection in such situations, adjustments to θ and computation of τ become necessary.

Using a heavy, sturdy tripod and wind-resistant casings can reduce unintentional camera movement. Camera noise depends highly on the used sensor as well as the lighting conditions. Smaller sensors usually exhibit higher noise levels than larger ones. Additionally, noise is amplified in low light conditions because the signal needs to be boosted. The noise value τ is estimated in a static scene by measuring the MSE over a specific duration and identifying the maximum value. This approach allows for the incorporation of both internal sensor noise and perturbations in the background, such as the motion of leaves on a bush, tree, or hedge, within the decision function. Another issue can be suddenly changing lighting conditions, such as when a room’s lights are turned on or off or when the sun suddenly appears or vanishes because of swiftly moving clouds. These circumstances will cause the barrier to be erroneously triggered. Pre-processing steps like intensity normalization, homomorphic filtering, or illumination modeling, as described by Radke et al. [34], can be used to solve this issue. This requires analysis of the entire image to provide reliable results. These possibilities were not explored due to their complexity exceeding the capabilities of a microcontroller. The color of the object is the final issue with the accuracy of the detection. Only when the background and the moving object to be detected are sufficiently distinct can a high MSE be achieved. Because grayscale images are much more prone to errors, using color images is advised whenever possible. For instance, the detection may not work if an athlete wearing a dark shirt moves in front of a dark background. The color of the athlete’s clothing is much less likely to match the background than the brightness.

#### 3.1.3. Multi-Barriers

To emulate the multi-barrier design of professional systems, several regions $\Omega_{1},\ldots ,\Omega_{m}$ can be defined for which the MSE is calculated. Then, the barrier is triggered only if the decision function (Equation 2) evaluates to 1 for all Ω regions. Since the MSE is invariant to the number of pixels in the region, the same θ can be used for all regions. The noise value τ should be computed independently. The rightmost image in Figure 1 shows an example of a multi-barrier scenario created with the proposed approach.

### 3.2. Mobile Image-Based Photoelectric Virtual Barrier System

#### 3.2.1. Hardware

We implement the image-based photoelectric barriers in a portable, useful, and economical system using the ESP32-Cam module [35]. A camera, microprocessor, and communication device are all included in this device. We developed an app for Android to configure, control and display the results. The ESP32-Cam and the necessary additional electronics and power source are placed in a simple casing and mounted on a tripod. Since Bluetooth is already natively supported by all devices, we don’t need any additional modules for wireless communication. The components needed for a single photoelectric barrier are listed in Table 2, along with their retail costs, including VAT and shipping (Germany, 2023). We can construct a fully functional photoelectric barrier for about 38 € in materials, which is already less than the cost of a single infrared photoelectric barrier sensor. The ESP32-Cam module, a breakout board with a 240 MHz ESP32 dual core System-on-a-Chip (SoC), a camera connector, and wireless connectivity like Bluetooth and WiFi, is at the heart of our system. As a result, it offers all the crucial features we need in a single, extremely affordable module. The majority of retailers include an OV2460 camera module, which we also use. The SoC includes 512 kB Ram and 4 MB flash memory, which is quite small for image processing and was the main bottleneck we had to consider designing the system. We added power supply and a status LED. Figure 2 depicts the wiring schematics.

The camera module is equipped with the OmniVision OV2640 1/4″ CMOS 2 Megapixel image sensor with rolling shutter technology, boasting a maximum resolution of 1620 × 1200 pixels (UXGA) at 15 fps, SVGA at 30 fps, and CIF at 60 fps with JPEG compression. A critical consideration is that JPEG decoding is handled in hardware, as the ESP32 has limitations in bandwidth, RAM, and processing power, making JPEG compression essential for performance. However, this compression poses a challenge for the implementation, as the evaluation should occur directly on the microcontroller and decompressing JPEG on the device is not feasible within an acceptable timeframe. To address this, a resolution of 160 × 240 pixels (HQVGA) and RAW data in RGB565 format are used. As a result, the number of images provided by the camera module is significantly reduced to about 26 per second. However, this approach enables the implementation of virtual barriers, fulfilling the project’s requirements.

#### 3.2.2. Communication and Clock Synchronization

Reliable device communication is a crucial component of the system. The best available option is Bluetooth because it works with all devices and has a sufficient range. We had no problems with line-of-sight distances of up to 50 m in an outdoor environment and with the default configuration of all devices. If the smartphone is positioned exactly in the center of the two sensors, this allows us to track distances of at least 100 m. A major issue is the variable latency of the Bluetooth connection, which ranges from a few to several hundred milliseconds and is extremely volatile. In order to obtain precise timings, the clocks must be synchronized. We decided to implement an online synchronization at setup time instead of a wired one.

From the smartphone, we periodically transmit “ping” messages to the photoelectric barrier device. The message initially contains the internal clock of the smartphone. The photoelectric barrier adds its own internal clock and sends the message back. Such a package gives us both the internal clock values of the two devices and the package’s latency. The distinctions between forward and backward latency are the only things that are uncertain. Repeating this procedure allows us to average out the uncertainty and obtain a precise clock synchronization. Then, in order to have time information in the same domain, we compute the clock offset value between the smartphone and the device clock. We perform this step every time the smartphone connects to the device. This makes the system easy to use, as there is no need for the user to perform a special synchronization operation.

#### 3.2.3. Android Control App

Since Android is the most popular mobile operating system and almost all smartphones and tablets meet the requirements for our system, we created an Android app to control and configure the photoelectric barrier devices. Young athletes and novice users alike can operate the system thanks to its simple and practical design. One button is all that is needed to start and stop the measurement once the barriers are connected and configured. Figure 2 shows screenshots of the app. The current camera image can be viewed in the configuration tab to align the virtual measurement line correctly with the real one. Multi-barriers can also be configured there.

#### 3.2.4. Power Supply and Operation Time

The ESP32 microcontroller is a power hungry device, especially with activated radio. With activated Bluetooth connection and video capture we measured about 300 mA at 3.3 V. So we can assume an average power consumption of one watt per hour. To power our mobile device, we use a rechargeable 9 V lithium-ion battery pack with a total capacity of 600 mAh according to the manufacturer’s datasheet. Assuming two battery cells in series with a nominal cell voltage of 3.6 volts each, we have 2 · 3.6 V · 0.6 Ah = 4.32 Wh. This provides at least 4 h of operation before the battery needs to be changed or recharged. In order to power the ESP32-Cam, we need a stable power supply of 3.3 V. We use an HW-411 (LM2596) DC-DC buck converter to convert the volatile battery voltage to the required 3.3 V.

Although the ESP32-Cam module has an internal voltage transformer and could be powered directly from the battery, it is much more efficient to use a separate transformer module. When powering the module this way, we measured 300 mA regardless of the voltage. We assume the device simply uses a Zener diode to cap the voltage to 3.3 V Assuming the nominal Li-ion cell voltage, the power consumption would rise to 2 · 3.6 V · 0.3 A = 2.16 W, and the theoretical operating time would be more than halved. Actual runtime will be even lower because the battery cannot reliably provide that much power until it is completely drained. Another issue is the massive amount of power that is converted to heat, which may require active cooling of the internal voltage regulator. Otherwise the unit could overheat and be destroyed. We use only passive cooling and have never experienced overheating problems with the proposed design.

#### 3.2.5. Reliability

To address the various reliability concerns, we measured the sensor noise and obtained an MSE of 20 to 30 depending on the lighting conditions. By default, we use a threshold of θ=400, which works reliably in most situations. Therefore, the sensor noise has only a marginal effect on the result and can be ignored in most cases. Our simple casing design and lightweight tripod are both susceptible to wind, but we only experienced problems with strong gusts. Vibrations of the floor, e.g., in a gym, did not affect the reliability of the system.

#### 3.2.6. Online Repository

We make the schematic, microcontroller program, and Android app publicly available under the MIT license so that anyone can build and use the proposed system. The required data will be published online on GitHub https://github.com/Tachikoma87/CBPhotoelectricBarrier (accessed on 22 August 2023). Figure 3 shows our implementation of one camera-based photoelectric barrier, as used in the experiments in the next section.

## 4. Experiments

In order to determine the accuracy of the novel system, two experiments timing 25-m flight runs using a professional TAG Heuer^®^ dual photoelectric barrier system (THS) and the novel camera-based system (CBS) have been conducted with a Samsung J3 (SM-J320F, Android 6.1.1, Suwon-si, Republic of Korea) as controlling device. Accuracy is evaluated using Bland-Altman plots [36].

In the experiment setup, one THS photoelectric sensor was positioned at a line on the round course of a track and field stadium, and the second one was placed 25 m away from the first sensor on a different line. The gate distance was approximately 1.5 m, and the sensors’ height was adjusted to activate when the athlete’s chest crossed the measurement line. To avoid sensor blockage, the CBS sensors were positioned below the THS sensors, and the CBS sensors were directed upwards to detect the athlete’s chest due to the lower vantage point. The measurement lines were aligned as closely as possible to resemble the configuration shown in the middle image of Figure 1 by visual inspection. This configuration is not flawless as the THS reliably triggers with the athlete’s chest, whereas the CBS may trigger with a limb. Also distance to the sensor, if athlete passes the gate more to the left or the right, may influence timing, since the CBS’s measurement line is not parallel to the track. Athletes initiated the sprint one meter before the first measurement line, and the reported timings of both systems were recorded for each trial.

Two experiments were conducted using the described setup on two separate days, involving experienced track and field athletes. The first group consisted of 14 children aged between ten and twelve. Each participant underwent six trials, with the initial two runs executed at a low speed, followed by two runs at a medium speed, and concluding with two runs at a high speed. In the second experiment, five adult track and field athletes participated, completing a total of nine trials each. The initial three trials were conducted at a slow speed, the subsequent three at a medium speed, and the final three at a high speed.

For each experiment, Bland-Altman plots were generated to compare the CBS with the THS, The plots are presented in Figure 4. In the first experiment, 79 out of 84 measurements (94.05%) fell within the 95% limits of agreement, while in the second experiment, 42 out of 45 measurement points (93.33%) were within the 95% limits of agreement. These results closely approach the expected 95% agreement within the limits and provide substantial evidence that the novel system performs with high accuracy similar to the professional system. The CBS, with a theoretical time resolution of 40 ms due to the microprocessor’s capability of evaluating up to 25 frames per second, achieved remarkably accurate timing for flying runs when compared to the THS’s rapid reaction time of 0.5 ms for the employed photocell HL-2-31.

The experimental results revealed an error of 94±62 ms for the children and −52±91 ms for the adults. The variability in the outcomes could potentially stem from the less than optimal setup of the two timing systems, as discussed earlier. Given the larger dataset available from the experiment with children, the accuracy of the proposed system is assumed to be 62 ms. This value serves as a reference for comparison with other available timing systems.

## 5. Discussion

This study conducted a comprehensive review of existing electronic timing systems designed for measuring sprint times in short distance runs. The findings highlighted the increasing utilization of handheld smart devices in training sessions for accurately assessing athletes’ performance, providing a cost-effective and reliable source of information. However, the review also unveiled a lack of research in this area and a missing interdisciplinary collaboration between computer science, engineering, and sports sciences. The review further emphasized that claims of timing system accuracy are often exaggerated, with accuracies exceeding 30 ms frequently lacking substantial evidence for short distance sprints. In response to these gaps, this research proposed an innovative approach to timing systems, utilizing low-cost cameras and a simple differencing method. The implementation utilized two 9 € breakout boards (ESP32-Cam) to create a complete system that emulates the functionality of photoelectric cells timing systems at a fraction of the cost (as shown in Table 1).

The prototype of this system exhibited an accuracy of 62 ms in the conducted experiments, placing it on par with other available electronic timing systems. This demonstrates that expensive and intricate timing systems can be effectively replaced by more cost-efficient and user-friendly alternatives during training sessions, without compromising accuracy. Notably, the proposed system can be assembled with materials costing less than 80 €, with a significant portion of the budget allocated to tripods. This renders it a budget-friendly alternative to professional systems, especially suitable for smaller sports clubs and schools that may have budget constraints. Furthermore, the system offers quicker deployment and simpler setup compared to commercial photoelectric cell timing systems since it doesn’t require the use of reflectors. Adhering to a hardware limitation of 25 frames per second, the system achieves a potential accuracy of only 40 ms (achieving 62 ms in experiments), which is well-suited for various applications including mass sports events, training sessions, and physical education in schools.

## 6. Future Work

It would be of interest to directly apply the proposed method to smart devices and evaluate its accuracy. However, initial testing by the authors suggests that smartphones may not be well-suited for such tasks, exhibiting notable limitations at the user level, particularly in simultaneously capturing and processing camera images. Similar applications like [18,19] demonstrate the practicality of the principle, yet their evaluation remains limited and often requires a high fps camera system to yield accurate results. Another avenue for research could involve investigating the accuracy of the proposed system with the multi-barrier setting. Leveraging the findings of Bond et al. [4], optimal height settings for the virtual barriers could be determined.

It is imperative to enhance interdisciplinary research collaboration between engineering, computer science, and sports science. This would pave the way for providing non-professional athletes and coaches with the means to accurately, reliably, and cost-effectively assess performance parameters. This research significantly contributes to achieving this objective by making the proposed timing system open source and available to the community.

## Figures and Tables

**Figure 1 sensors-23-07339-f001:**
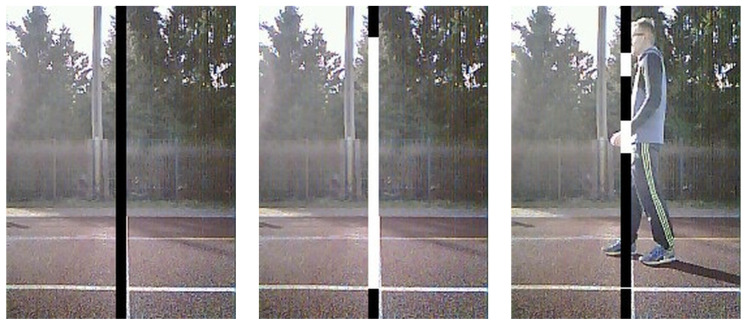
Camera footage with the detection area shown in black and the white areas marking regions, where the image-to-image error is actually calculated. **Left**: Full virtual barrier. **Middle**: Default virtual barrier in white. **Right**: Multi-barrier setting.

**Figure 2 sensors-23-07339-f002:**
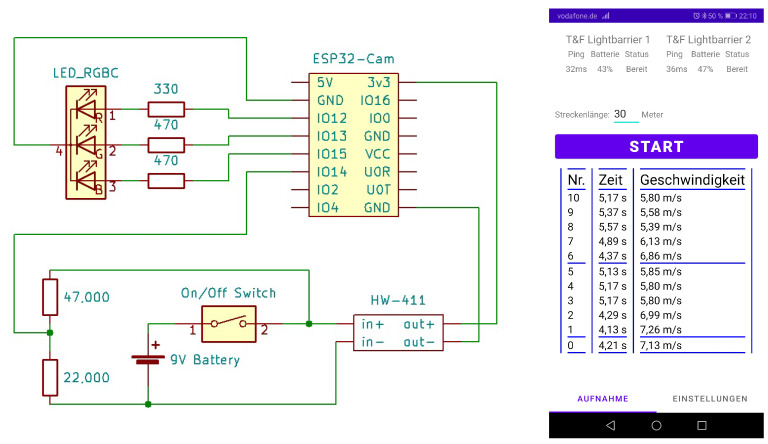
The left image of this figure shows the schematics for one photoelectric barrier. The RGB diode is used to show the status of the device and the voltage divider is used to measure the battery charge. The right image shows a screenshot of the control app (German version), which displays important parameters of the hardware and the recorded trials. Our app is compatible with all Android smartphones and tablets version 5.0 and above that have Bluetooth capabilities.

**Figure 3 sensors-23-07339-f003:**
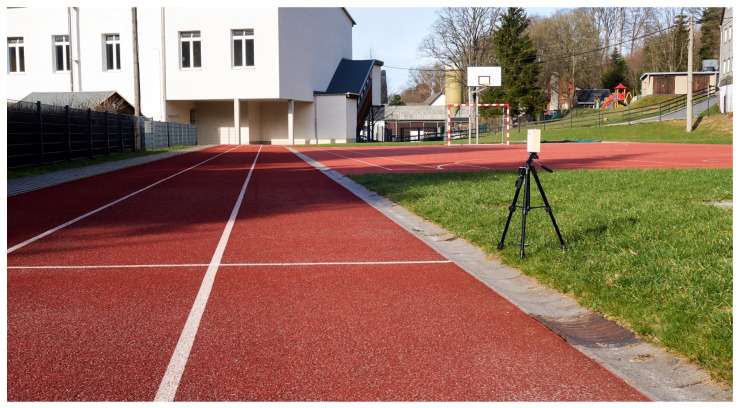
Our camera-based photoelectric barrier can be easily placed on a tripod. Unlike traditional systems, no reflector is required, which greatly reduces setup time. Bluetooth communication eliminates the need for wires.

**Figure 4 sensors-23-07339-f004:**
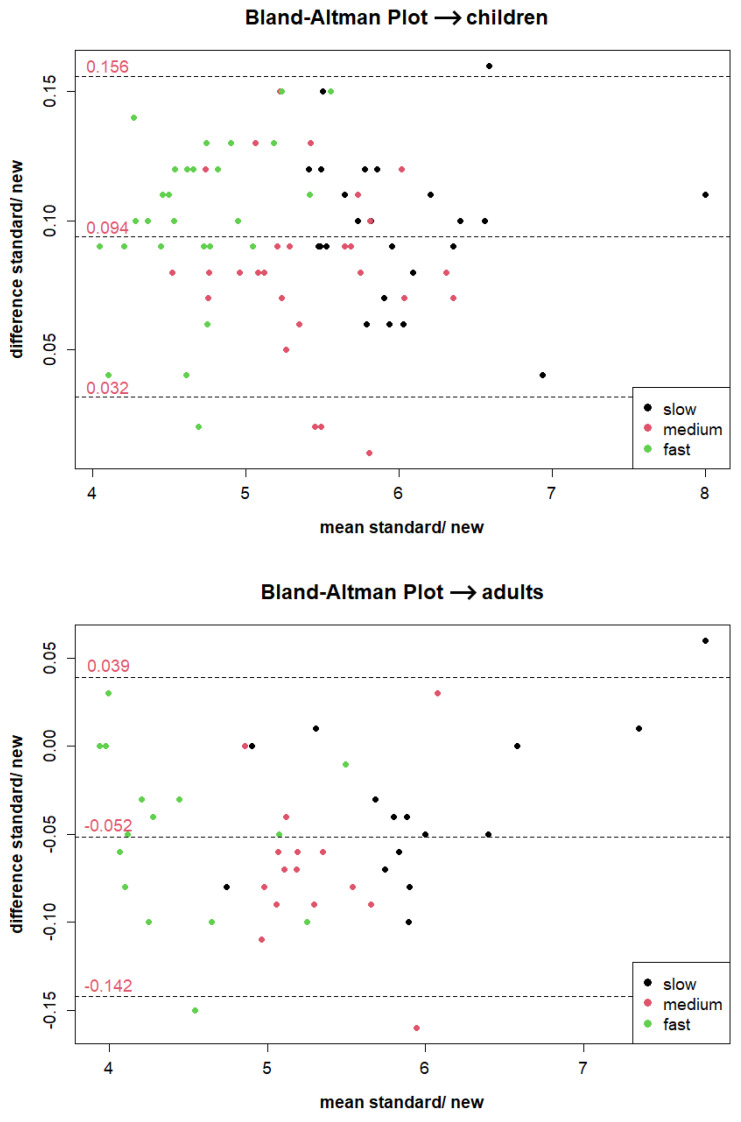
Bland-Altman plots of the two experiments performed. Almost all measurement points are within the 95% limits of agreement, which shows that the proposed system has similar accuracy to a professional photoelectric barrier system.

**Table 1 sensors-23-07339-t001:** There are numerous commercial systems available for timing short distance sprints. This list provides an overview of exemplary devices for each technology currently in use. The prices mentioned are intended to convey the magnitude of the initial cost and reflect the snapshot as of August 2023. The accuracies are derived from the results of studies found in scientific literature, whenever available. Additionally, the overview includes information about whether each system requires an additional smart device for its operation.

Device	Price (€)	Technology	Smart Device	Accuracy
Stopwatch	Free	Manual	Android/iOS	350 ms [2]
Cronox 3.0—WIRELESS [3]	500	RPC	Android/iOS	30 ms [4]
Witty wireless [5]	2400	RPC	no	30 ms [4]
ks-sport wireless [6]	1712	RPC	no	30 ms [4]
SmarTracks [7]	n.a.	Magnetic	Android/iOS	77 ms [8]
Finishlynx RFID [9]	6700	RFID	no	45 ms [10]
StatSports [11]	280	GPS + IMU	Web-Interface	n.a.
Catapult One [12]	180/year	GPS + IMU	Web-Interface	n.a.
Radar Gun [13]	220	Radar	no	2 km/h [14]
FAT Photo-Finish [15]	>10,000	Camera	no	1 ms
My Sprint [16]	10	Camera	iOS	28 ms [17]
SprintTimer [18]	6+	Camera	iOS	10 ms
Photo Finish [19]	100	Camera	2 × Android	10 ms
Proposed device	72	Camera	Android	62 ms

**Table 2 sensors-23-07339-t002:** Component list for a single camera-based photoelectric barrier. These are the prices in online stores in Germany including VAT and shipping in 2023. Materials such as wires, casing materials, and solder are summarized as miscellaneous. A high-capacity rechargeable 9 V lithium-ion battery is used to ensure sufficient operating time.

Component	Price
Tripod	20 €
ESP32-Cam	9 €
Battery (Lithium Ion)	6 €
Buck Converter	1.50 €
Miscellaneous	1.50 €
Total	38 €

## Data Availability

Software and schematics necessary to reproduce our proposed system are available under MIT license on Github: https://github.com/Tachikoma87/CBPhotoelectricBarrier (accessed on 22 August 2023). The data presented in this study are available on request from the corresponding author.

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
