# Peer review of "A Novel Inexpensive Camera-Based Photoelectric Barrier System for Accurate Flying Sprint Time Measurement"

_sensors, 2023, doi:10.3390/s23177339_

Round 1
Reviewer 1 Report
Dear authors,
The topic & way the authors presented the work is good, but here i've few suggestions to the authors to improve the quality of the manuscript.
1) Without any Comparative table, how you state the device as inexpensive?
2) The introduction needs to be improved. It should focus on the meaning of this study, and the aims of all the literature reviews should lead to the research object.
3) the authors need to provide a comprehensive explanation of how the device was developed, the paper does not provide a clear description of the testing phase of the device, which is crucial information for the readers to evaluate the validity of the results. \
4) The words in Bland-Altman plots are too small.
5) Line-129 our method’s accuracy is independent of the smartphone being used. Which are the phones on which your system as been tested? What is the configuration?
Nothing, small amount of spell check is required
Reviewer 2 Report
The comments are in the PDF document.

The English Language of the whole paper should be corrected!
Round 2
Reviewer 2 Report
The paper is improved, thank you.